# Rethinking Prompt Design for Inference-time Scaling in Text-to-Visual Generation

## Abstract

Achieving precise alignment between user intent and generated visuals remains a central challenge in text-to-visual generation, as a single attempt often fails to produce the desired output. To handle this, prior approaches mainly scale the visual generation process (e.g., increasing sampling steps or seeds), but this quickly leads to a quality plateau. We argue that this limitation arises because the prompt, crucial for guiding generation, is kept fixed. To address this, we propose Prompt Redesign for Inference-time Scaling, coined PRIS, a framework that adaptively revises the prompt during inference in response to the scaled visual generations. The core idea of PRIS is to review the generated visuals, identify recurring failure patterns across visuals, and redesign the prompt accordingly before regenerating the visuals with the revised prompt. To provide precise alignment feedback for prompt revision, we introduce a new verifier, element-level factual correction, which evaluates the alignment between prompt attributes and generated visuals at a fine-grained level, achieving more accurate and interpretable assessments than holistic measures. Extensive experiments on both text-to-image and text-to-video benchmarks demonstrate the effectiveness of our approach, including a 15% improvement on VBench 2.0. These results highlight that jointly scaling prompts and visuals is key to fully leveraging scaling laws at inference-time.

## 1 Introduction

Generative models (Comanici et al., 2025; Labs, 2024; Wan et al., 2025) have achieved remarkable progress across various domains, including language, image, and video domains, demonstrating strong capabilities in modeling complex data distributions. In the visual domain, denoising models (Ho et al., 2020; Lipman et al., 2023) conditioned on textual prompts now allow users to generate high-quality images and videos directly from natural language descriptions. However, as prompts become more intricate, such as requiring compositional structures in images or motion dynamics, camera movements, and causal relationships in videos, it becomes increasingly challenging to produce outputs that fully align with the prompt in a single attempt.

Recent work addresses this shortfall in text-visual alignment by allocating additional compute at inference time (i.e., inference-time scaling). These approaches typically scale the visual generation either by increasing the compute budget for decoding a single candidate from a prompt (Ma et al., 2025), or by generating multiple candidates for the same prompt to produce a diverse pool of visual outputs (Kim et al., 2025a;b; He et al., 2025). However, they primarily focus on scaling visual parts while keeping the input prompt fixed. This creates a key bottleneck because many generation errors arise from ambiguous or incomplete prompts, and scaling visuals conditioned on a suboptimal prompt offers limited benefit since the prompt provides essential guidance for conditional generation.

In parallel, another line of work (Brade et al., 2023; Wang et al., 2024b; Datta et al., 2024; Hao et al., 2023) focuses on ensuring that the model interprets prompts in alignment with the user's intent. These methods typically rewrite the user input prompt to produce model-preferential outputs or enable prompt exploration at inference time based on a single output. While such refinements improve how well a generative model interprets text, they operate solely in the text domain, and adjustments are confined to individual samples. In other words, these approaches do not adapt the prompt in conjunction with visual scaling, where recurring generative failures or consistent patterns emerge

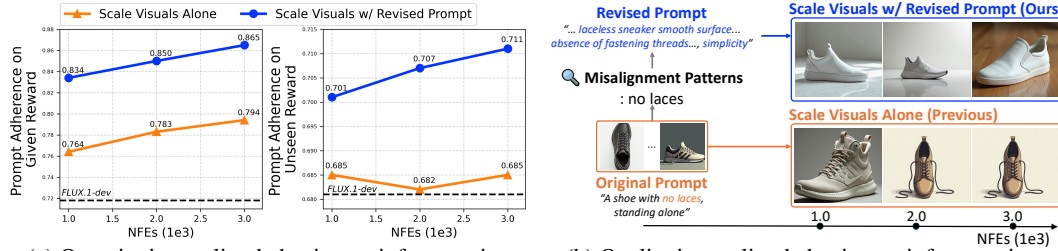

(a) Quantitative scaling behaviors at inference time     (b) Qualitative scaling behaviors at inference time

Figure 1: **Our prompt redesign scales with compute, while fixed-prompts plateau**. Given a user-provided text prompt, scaling visuals alone with a fixed prompt at inference time often leads to early performance plateaus, especially for unseen rewards (see orange lines), and repeatedly produces outputs with only partial prompt coverage even as compute increases. In contrast, scaling visuals alongside our redesigned prompts yields progressively improved generations and substantially higher prompt-adherence scores as compute increases for both given and unseen rewards (see blue lines).

when scaling visual outputs. As a result, they miss the opportunity to jointly scale both the prompt and the visuals for improved text-visual alignment.

**Contributions.** To address these limitations, we extend inference-time scaling beyond visuals to the text prompt, proposing Prompt Redesign for Inference-time Scaling (PRIS). PRIS diagnoses early failures and injects fine-grained feedback into subsequent generations to revise prompts. Instead of passively waiting for a high-scoring sample, PRIS identifies recurring failure patterns across generated visuals and adaptively revises the prompt to emphasize the under-addressed aspects while preserving the original intent. By doing so, in contrast to fixed-prompt inference-time scaling, which quickly plateaus in prompt adherence, even with more compute (see orange line in Figure 1a), PRIS leverages compute effectively by scaling prompts and visuals jointly, thereby following the scaling law and achieving sustained improvements in text-visual alignment (see blue line in Figure 1a).

To identify failure patterns for prompt revision, PRIS relies on fine-grained verification of each visual. For this purpose, we develop Element-level Factual Correction (EFC), an interpretable and descriptive verifier to examine the generated visuals, built on a multimodal large language model (MLLM) (see Figure 2). When assessing the alignment between the visuals and the prompt, EFC first decomposes the prompt into disjoint semantic elements and verifies each against a caption of the generated visual, framing every element as a textual hypothesis. This text-to-text comparison mitigates the affirmative bias common in MLLM-based text-visual question answering (Fu et al., 2025; Bai et al., 2024; Han et al., 2024), thus improving verification accuracy. We further introduce a benchmark pairing each prompt with multiple generated visuals, some fully aligned, others only partially so. On this dataset, EFC consistently distinguishes ground-truth visuals from plausible but misaligned distractors, significantly outperforming prior alignment-measuring verifiers.

Building on our new verifier, we conduct extensive experiments showing that EFC-guided prompt redesign, PRIS, is key to leveraging scaling laws. Our approach consistently improves text-visual alignment without sacrificing visual quality, achieving a 7% gain on GenAI-Bench for text-to-image and a 15% gain on VBench2.0 for text-to-video, by identifying recurring failure patterns and refining prompts accordingly. In addition, since PRIS is complementary to existing inference time scaling methods, which mainly focus on scaling visuals, integrating our approach further enhances text-visual alignment. Moreover, the strong performance of EFC underscores its broader potential as a versatile and interpretable tool for selecting text-aligned visual outputs, and our benchmark further provides the first systematic evaluation of verifiers for inference-time scaling, enabling direct measurement of their ability to detect fine-grained misalignment that limits scaling efficiency.

## 2 RELATED WORK

**Scaling inference-time compute in visual generation.** Despite recent progress driven by powerful denoising architectures (Ho et al., 2020; Lipman et al., 2023), producing faithful outputs in text-to-visual generation remains challenging, particularly for complex prompts. Since outputs from these models are determined jointly by the initial noise, the sampling trajectory, and the prompt, this motivates inference-time scaling methods that allocate additional compute to exploring favorable

noise seeds and trajectories. Typical strategies include increasing decoding steps for a single sample to improve quality, or generating multiple candidates (Ma et al., 2025), often with advanced algorithms (Kim et al., 2025a;b) that expand the search space. Here, learned reward models (Liu et al., 2025a; Zhang et al., 2025) provide a scalar alignment score for each sample and serve as verifiers that select the best among $N$ generated samples (Best-of-$N$; BoN). Selection can be performed either on the final outputs, as in BoN, or during sampling, as in Search-over-Paths (Kim et al., 2025a;b; He et al., 2025), which iteratively resamples and propagates high-reward candidates along the denoising trajectory and often outperforms naïve BoN since high-reward outputs cluster in local regions of the sample space. However, all of these methods operate under a fixed prompt, expanding only the visual search space and discarding earlier low-scoring generations. We take a different view: rather than treating these earlier generations as expendable, we revisit and analyze them, and more importantly, redesign the prompt alongside visual scaling to provide stronger guidance for subsequent generations.

**Prompt design in text-to-visual generations.** Prompt design is a critical component of text-conditioned generation that serves not merely as a pre-processing step (Zheng et al., 2025) but as a means to improve model comprehension, output quality, and adherence to the input description. Since the prompt itself guides the generation, even different phrasings of the same user intent can produce markedly different outputs. Yet, crafting effective prompts remains challenging, often requiring tedious trial-and-error. To address this, recent approaches (Brade et al., 2023; Wang et al., 2024b; Datta et al., 2024; Hao et al., 2023; Hei et al., 2024) propose systems that interactively help users explore alternative phrasings or automatically rewrite prompts, reducing reliance on naïve iterations. These methods, however, depend on human involvement (Brade et al., 2023; Wang et al., 2024b) or do not explicitly target text adherence in the context of inference-time scaling (Datta et al., 2024; Hao et al., 2023; Hei et al., 2024). This gap is critical: scaling visuals alone often reproduces recurring failure patterns without improving adherence, even with more compute. This highlights the need for prompt refinement strategies that address failures across samples, rather than noisy per-sample revisions. Therefore, in this work, we design prompts specifically for inference-time scaling with the goal of increasing adherence as compute grows. Our method applies to both T2I and T2V generation, extending beyond prior work that has focused primarily on T2I generations.

**Chain-of-thought and reasoning.** Incorporating chain-of-thought (CoT) reasoning into visual generation has emerged as a promising paradigm for improving image quality through iterative reflection and guidance (Wang et al., 2025a; Jiang et al., 2025; Liao et al., 2025; Guo et al., 2025; Zhuo et al., 2025). Recent works pursue this direction via unified models (Tian et al., 2025) that combine visual understanding and generation and jointly optimize large language models with multimodal objectives and generation-specific losses. In contrast, we integrate off-the-shelf MLLMs, without additional training, into existing visual generation pipelines. Unlike unified models, these visual generators lack reasoning capabilities to refine themselves for prompt planning and reflective refinement. Thus, the central challenge we address is how to bridge MLLM reasoning with visual generative models to enable diagnostic feedback and effective intervention.

## 3 PRIS: PROMPT REDESIGN FOR INFERENCE-TIME SCALING

This section begins with our key observations that motivate the explicit consideration of prompts with visual feedback in inference-time scaling (Section 3.1). Next, we introduce our new verifier, which provides fine-grained feedback on generated visuals (Section 3.2), and finally present our verifier-guided prompt redesign strategy to improve prompt fidelity in inference-time scaling (Section 3.3).

### 3.1 MOTIVATION FOR PROMPT ADJUSTMENT IN INFERENCE-TIME SCALING

Prior inference-time scaling methods treat noise and trajectory as the primary levers of search, keeping the prompt fixed. However, the prompt is an equally critical determinant of the final output and remains underexplored in inference-time scaling. This raises the question: *can the prompt be adapted during inference to better align with user intent*? We observe that generated samples reveal visual misalignment patterns: some elements are consistently satisfied, while others are consistently missed. For example, in Figure 1b, when scaling with the intent "a shoe with no laces, standing alone," the element "a shoe" is consistently achieved, yet laces still appear in every output. See Appendix A for broader examples across image and video prompts. These observations motivate revising the prompt to target recurring misalignments rather than blindly scaling visuals under a suboptimal prompt.

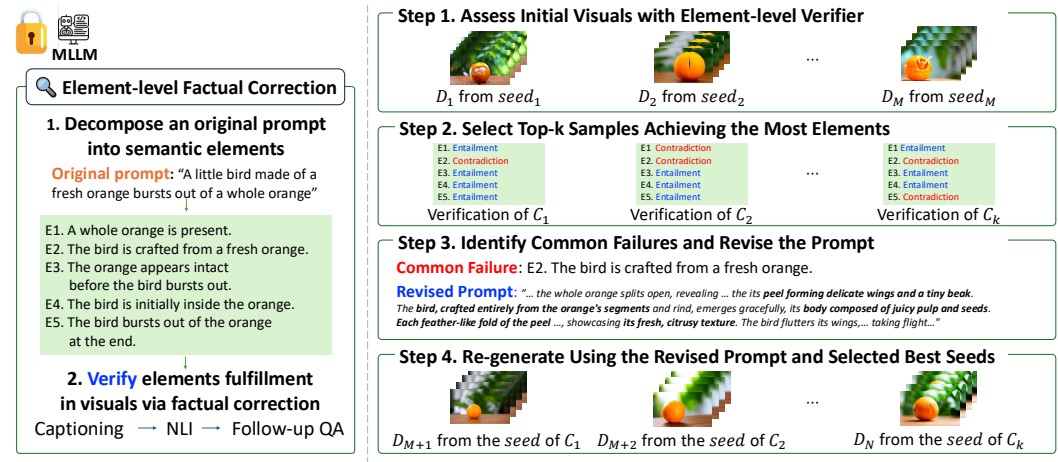

(a) Element-level Factual Correction (EFC) | (b) Prompt Redesign at Inference-time Scaling (PRIS)

Figure 2: **Overview of Prompt Redesign for Inference-time Scaling (PRIS)**, which leverages diagnostic feedback from our verifier EFC to revise prompts during inference based on generated visuals. EFC decomposes prompts into semantic elements and verifies each element for fine-grained text-visual alignment (left). Guided by the EFC, PRIS proceeds as follows (right): Step 1 reviews initial generations with EFC; Step 2 selects top-$k$ successful samples and identifies recurring failures; Step 3 redesigns the prompt to emphasize common failures; and Step 4 regenerates visuals with the revised prompt and top-$k$ seeds. The process can be iterated by returning from Step 4 to Step 2.

## 3.2 ELEMENT-LEVEL ALIGNMENT ASSESSMENT BETWEEN PROMPT AND VISUALS

Our goal is to identify recurring misalignments between the original prompt and generated visuals, such as missing elements, incorrect causal relations, or disordered temporal motions, by reviewing the current set of generations. This requires a fine-grained visual verifier that, for each generated visual, evaluates which prompt attributes are satisfied or missed, since single-scalar alignment scores from previous verifiers (Lin et al., 2024; Liu et al., 2025a) cannot reveal such detail. To this end, we introduce Element-level Factual Correction (EFC), a new verifier that provides fine-grained and interpretable text-visual assessments using an MLLM without additional training. See Figure 2 (a) for an overview; further illustrations of EFC are provided in Appendix B.

**Break down the prompt for precise element-level assessment.** Holistic evaluation of prompt alignment often obscures precise verification. As prompts grow more complex and nuanced, it becomes increasingly difficult to determine which attributes are properly addressed and which are overlooked. To address this, EFC first decomposes the original prompt $p$ into a set of verifiable atomic semantic elements $p = \{p_1, p_2, ..., p_s\}$, where each element $p_i$ corresponds to a distinct element. Here, atomic facts are extracted according to predefined semantic categories, such as image-level elements covering object presence, properties, and spatial arrangement, and motion-level elements covering object motion, camera movement, scene transitions, and temporal ordering. Then, EFC classifies each $p_i$ as either {core, extra}. The core elements are objective, factual, and essential to the intent of the prompt, while the extra elements are more subjective or stylistic, so they are often flexibly interpreted. This classification is later used to prioritize generated samples during final scoring.

**Factual correction to precisely assess element-level alignment.** After decomposing the prompt into multiple elements, EFC performs factual correction on each element by evaluating it against each generated visual $D$ to determine whether the element is accurately realized. Instead of direct visual question answering (VQA), EFC first generates a caption for $D$ in natural language, then infers the relationship between each element $p_i$ and the caption of $D$. This step is formulated as a natural language inference (NLI) task: if the caption semantically supports the element, the relationship is labeled as *entailment*; if the caption contradicts the element, it is labeled as *contradiction*; and if the caption does not provide sufficient information to confirm the element, the label is *neutral*.

For any element $p_i$ initially classified as *neutral*, such as when the caption omits or ambiguously describes it, EFC generates an open-ended question $q_i$ that restates $p_i$ without binary framing and queries the question $q_i$ with the visual $D$. The free-form response is then re-evaluated against $p_i$ in a second NLI step, and the element is relabeled as either *entailment* or *contradiction*. We propose a

Table 1: **Quantitative results of T2I on GenAI-Bench.** * denotes results with standard prompt expansion; BoN denotes "Best of N" visual samples. **Bold** shows the best.

| Method | VQA-Score (Given) | DA-Score w. BLIP2-VQA (Unseen) | Aesthetic Quality (Unseen) |
|---|---|---|---|
| FLUX.1-dev | 0.718 | 0.681 | 5.764 |
| +BoN | 0.783 | 0.682 | 5.761 |
| +**PRIS** | **0.854** | **0.707** | **5.765** |
| FLUX.1-dev* | 0.769 | 0.695 | 5.824 |
| +BoN* | 0.829 | 0.710 | 5.820 |
| +**PRIS*** | **0.853** | **0.713** | **5.841** |

Figure 3: **Qualitative comparisons of T2I generation.** * denotes results with standard prompt expansion.

text-text verification approach, rather than direct yes/no VQA, because it achieves higher accuracy by mitigating confirmation bias and consistently providing interpretable descriptions, whereas binary VQA often omit such information. Detailed ablations are reported in Section 4.3 and Appendix B.

**Prioritize core elements in final scoring.** After factual correction, we obtain verification results $C$ for each visual. EFC then assigns a score based on the number of elements labeled as *entailment* in $C$. Core elements are prioritized because they are objective, factual, and less open to subjective interpretation, making them essential for faithfully capturing the prompt's intent. If multiple candidates achieve the same core accuracy, extra elements are used to further determine the ranking.

### 3.3 EFC-GUIDED PROMPT REDESIGN FOR INFERENCE-TIME SCALING

We propose a prompt redesign framework, PRIS, which revises the prompt using feedback from our verifier EFC, which provides fine-grained and interpretable assessments of text-visual alignment. By pinpointing where alignment breaks down in earlier visual outputs, PRIS incorporates diagnostic signals into subsequent generations, guiding them toward higher fidelity to the prompt.

- **Step 1. Generation and verification.** PRIS first generates $M$ candidate visual samples and evaluates the fulfillment of elements $\{p_1, p_2, \ldots, p_s\}$ for each sample using our verifier EFC (Section 3.2), obtaining verification results for each sample ($C_1$ through $C_M$).

- **Step 2. Select the top-$k$ best-performing samples.** PRIS then selects the top-$k$ samples that collectively cover the largest number of elements, with ties resolved using the scalar score from a learned reward model (Lin et al., 2024; Liu et al., 2025a) trained on human-preference datasets. This ensures that the selected candidates better reflect human-preferred outputs.

- **Step 3. Identify misalignment patterns and revise the prompt.** Within the selected subset, PRIS identifies common failures, defined as elements whose success probability is below 50% within the top-$k$ samples. Based on feedback from EFC about common failures, PRIS revises the original prompt $p$ into a revised prompt $p'$ that explicitly reinforces overlooked elements while preserving those already well addressed. This targeted refinement encourages subsequent generations to focus more effectively on underrepresented elements. If no common failures are observed (i.e., every element has a success probability above 50% within the subset), we instead treat the prompt itself as the refinement target to encourage exploration of prompt variations.

- **Step 4. Regenerate with the revised prompt and selected noise conditions.** Using the revised prompt $p'$, we regenerate $N - M$ samples initialized from the noise latents (seeds) of the top-$k$ samples. If $k < (N - M)$, PRIS produce multiple revised prompts by varying their phrasing. Since certain noise conditions yield better alignment for specific prompt types (Zhou et al., 2025; Ahn et al., 2024; Qi et al., 2024), reusing these seeds better preserves earlier successes than random initialization. After regeneration, we verify and rank the samples with our verifier EFC.

While the generation-prompt revision-regeneration loop can be repeated, in our main experiments we apply it once, as this already provides sufficient gains; further analysis is presented in Section 4.4. Within PRIS, EFC-guided prompt redesign leverages prior failures to refine prompts and exploit favorable noise configurations. By treating partially correct generations as informative feedback rather than discarding them, PRIS makes more effective use of generator compute and improves output fidelity, thereby bridging the reasoning capabilities of MLLMs and visual generation.

Table 2: **Quantitative comparisons of T2V generation on VBench-2.0.** * denotes results obtained using the standard prompt expansion, and **bold** indicates the best results. We use $N = 20$ samples for Wan2.1-1.3B (small) and $N = 10$ for Wan2.1-14B (large), which can lead to the smaller model achieving higher scores due to the larger number of samples.

| Category | Method | Dynamic Spatial Relationship | Dynamic Attribute | Motion Order Understanding | Human Interaction | Composition | **Average** |
|---|---|---|---|---|---|---|---|
| Controllability & Creativity | Wan2.1-1.3B* | 35.56 | 46.67 | 52.87 | 74.44 | 48.33 | 51.57 |
| | +BoN* ($N = 20$) | **43.33** | 53.33 | 51.72 | **90.00** | 50.2 | 57.73 ↑ +6.16 |
| | +**PRIS*** ($N = 20$) | **43.33** | **73.33** | **68.97** | **90.00** | **51.6** | **65.45** ↑ +13.88 |
| | Wan2.1-14B* | 50.00 | 48.89 | 43.33 | 78.89 | 47.18 | 53.66 |
| | +BoN* ($N = 10$) | 46.67 | 56.67 | 60.00 | 80.00 | 49.23 | 58.51 ↑ +4.85 |
| | +**PRIS*** ($N = 10$) | **60.00** | **73.33** | **66.67** | **90.00** | **54.23** | **68.85** ↑ +15.19 |

| Category | Method | Camera Motion | Motion Rationality | Mechanics | Material | Thermotics | **Average** |
|---|---|---|---|---|---|---|---|
| Commonsense & Physics | Wan2.1-1.3B* | 41.38 | 38.10 | 75.00 | 75.38 | **86.25** | 63.22 |
| | +BoN* ($N = 20$) | 37.93 | 35.71 | 84.00 | 73.91 | 81.48 | 62.61 ↓ −0.61 |
| | +**PRIS*** ($N = 20$) | **51.72** | **50.00** | 80.00 | **78.26** | 70.37 | **66.07** ↑ +3.46 |
| | Wan2.1-14B* | 36.67 | 40.00 | 83.33 | 77.78 | **79.49** | 63.45 |
| | +BoN* ($N = 10$) | **43.33** | 43.33 | **86.36** | 80.77 | 76.92 | 66.14 ↑ +2.69 |
| | +**PRIS*** ($N = 10$) | **43.33** | **53.33** | **86.36** | **88.46** | 76.92 | **69.98** ↑ +6.53 |

Original short prompt: *"A person is turning on the desk lamp."*
Initially expanded prompt: *"A person is... gently turning on a desk lamp... twist the lamp's switch... newly lit lamp casts warm light..."*
Revised prompt: *"A young... **hand resting gently on the base of a desk lamp**... the person twists the switch, and **the warm glow gradually illuminates the space**... **as the lamp turns on**... background remains softly blurred... **the transition from darkness to light emphasizes** the calming effect... warmly lit room..."*

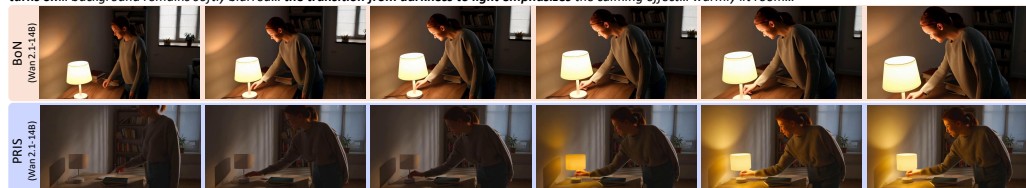

Figure 4: **Quantitative comparisons on T2V generation.** Our revised prompt elaborates on previous failures by emphasizing causal order, ensuring the lamp turns on immediately when touched.

## 4 EXPERIMENTS

We comprehensively evaluate PRIS and EFC for inference-time scaling. First, we study the effect of prompt redesign under a fixed compute budget (Section 4.1). Next, we analyze the scaling behavior of PRIS by expanding the generator's compute budget or iteratively revising prompts, and examine its integration with visual scaling algorithms originally designed for fixed prompts (Section 4.2). Finally, we assess the ability of EFC and existing verifiers to select the best-quality sample from mid-quality candidates (Section 4.3) and conduct ablations on both PRIS and EFC (Section 4.4).

### 4.1 EFFECT OF PRIS ON INFERENCE-TIME ALIGNMENT AND VISUAL QUALITY

We study the effect of prompt redesign on output quality under a fixed compute budget, defined as the number of function evaluations (NFE). For detailed experimental setups, please refer to Appendix C.

**Experimental setup.** For the MLLM verifier, we use Qwen2.5-VL (Bai et al., 2025) with prompt instructions for each process described in Section 4.3, without additional training. We compare with Best-of-N (BoN) (Ma et al., 2025), which generates $N$ samples at once and selects the best, while our method generates half of them (setting $M = \lfloor N/2 \rfloor$), revises the prompt with feedback, and regenerates the rest using the revised prompt and top-$k$ seeds. We set $k = \lceil N/4 \rceil$, thereby producing two revised prompt variants for the remaining $N - M$ samples. We also include standard prompt expansion (Wan et al., 2025) (denoted as *) to contrast with our failure-aware revisions. For generator selection, we first measure base fidelity using embedding similarity between the generated caption and the original prompt (Wang et al., 2023), retaining only models with sufficiently high scores and excluding those with substantially weaker prompt adherence.

For T2I generation, we use FLUX.1-dev (Labs et al., 2025) on GenAI-Bench (Li et al., 2024) with 320 sampled prompts (20% of the total) to avoid redundancy. For the guidance reward, we use VQA-Score (Lin et al., 2024). For held-out evaluation, we use DA-Score (Singh & Zheng, 2023) to assess fine-grained prompt adherence, and LAION (2024) to evaluate the aesthetic quality of the generated images. NFE is set to 2000 ($N = 20$, 50 denoising steps, classifier-free guidance (Ho &

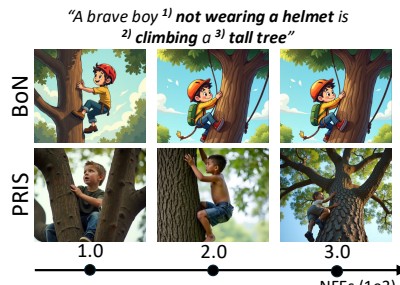

*"A brave boy 1) not wearing a helmet is 2) climbing a 3) tall tree"*

1.0  2.0  3.0

NFEs (1e3)

**Figure 5: Qualitative examples with increasing inference-time compute.** PRIS generates progressively taller trees while satisfying all attributes, whereas BoN consistently misses some.

Table 3: **Quantitative results for iterative prompt refinement with increasing inference-time compute.** Iteratively revision prompts consistently improves reward scores by addressing common failures, and the gains even generalize to unseen rewards. In contrast, fixed prompts often saturate and fail to transfer.

| Method | NFEs (1e3) | VQA-Score (Given) | DA-Score w. BLIP2-VQA (Unseen) | Aesthetic Quality (Unseen) |
|---|---|---|---|---|
| Initial (w.o. revision) | 0.5 | 0.736 | 0.679 | 5.756 |
| Initial (w.o. revision) | 1.0 | 0.764 | 0.684 | 5.766 |
| 1st revision | 1.0 | 0.834 | 0.703 | 5.755 |
| Initial (w.o. revision) | 1.5 | 0.776 | 0.683 | 5.751 |
| 2nd revision | 1.5 | 0.849 | 0.705 | 5.740 |

Salimans, 2022)=3.5). For T2V, we use Wan2.1-1.3B/14B (Wan et al., 2025) with VideoAlign (Liu et al., 2025a) as guidance, and evaluate on VBench2.0 (Zheng et al., 2025) across four dimensions: controllability, creativity, commonsense, and physical plausibility. NFE is set to 2000 ($N = 20$, 50 steps, cfg=6) for Wan2.1-1.3B and 1000 ($N = 10$, 50 steps, cfg=6) for Wan2.1-14B.

**Experimental results on T2I generation.** We present results in Table 1, Figure 3, and Appendix D. As shown in Table 1, our approach PRIS consistently outperforms all baselines across metrics. Notably, it yields substantial gains in prompt adherence while maintaining comparable aesthetic quality. Even against the standard prompt expansion variant (denoted as *), our method achieves significantly higher scores. These results suggest that prompt expansion is most effective when guided by visual feedback, rather than by adding arbitrary details as in standard prompt expansion. The qualitative results in Figure 3 further support this claim, showing that PRIS exhibits a stronger ability to handle complex, compositional prompts compared to BoN. For the top row in Figure 3, after identifying layout specification as a challenge in the initial outputs, our method revises the prompt to emphasize layout-related details. Likewise, for the prompt "fork not made of wood" (bottom row in Figure 3), where BoN still produces wooden forks due to the negation, our method explicitly instructs the model to generate "silver forks," thereby resolving the misunderstanding.

**Experimental results on T2V generation.** Our method delivers substantial improvements in prompt alignment for T2V generation, as shown in Table 2, Figure 4, and further examples in Appendix D. PRIS achieves gains of +13.88% and +15.19% in the Controllability and Creativity categories for the small and large models, respectively. This significantly surpasses BoN*, which applies standard prompt expansion at initialization without visual feedback on where to focus. Specifically, the largest gains appear in Dynamic Attribute and Motion Order Understanding, which require sequential reasoning (e.g., "A then B," "A transitioned to B"). Here, PRIS identifies failures in the initial outputs and revises prompts to clarify how sequences should unfold, emphasizing the parts that previously failed. Qualitative examples, including revised prompt in Figure 4, illustrate these improvements. Beyond these categories, PRIS also improves Commonsense and Physics by +3.46% and +6.53%, respectively. A notable exception is Thermotics, where performance drops slightly due to the reward model overfitting to exact numeric values rather than broader physical plausibility. Finally, while Zheng et al. (2025) suggests that camera motion is largely determined by base model capacity, our results show that refining prompts to specify how camera motion should unfold in conjunction with other scene elements can still yield measurable improvements.

## 4.2 SCALING BEHAVIORS OF PRIS

**PRIS scales with increasing NFEs.** We provide both quantitative and qualitative evidence that PRIS scales with increasing NFEs, whereas a fixed prompt quickly saturates and fails to scale (Figures 1 and 5). In Figure 1a, BoN, which relies on fixed prompts, shows monotonic gains in guidance (seen) rewards, but its held-out performance saturates beyond 1e3 NFEs. In contrast, PRIS continues to improve, achieving higher accuracy even under held-out evaluation. Figures 1b and 5 confirm this trend qualitatively. In Figure 1b, BoN repeatedly generates "shoe with laces," whereas PRIS revises the prompt to realize "shoe without laces," improving alignment as compute increases. Similarly, in Figure 5, our method produces progressively taller trees while satisfying all attributes, even at smaller budgets, whereas BoN repeatedly fails to follow the prompt by generating a boy wearing a helmet.

Table 4: **Quantitative results of integrating PRIS with T2I visual scaling methods** on GenAI-Bench. **Bold** shows the best.

| Method | VQA-Score (Given) | DA-Score w. BLIP2-VQA (Unseen) | Asthetic Quality (Unseen) |
|---|---|---|---|
| SDXL | 0.639 | 0.652 | 5.759 |
| +BoN | 0.649 | 0.663 | 5.810 |
| +DAS | 0.657 | 0.671 | 5.819 |
| +DAS w/ **PRIS** | **0.700** | **0.688** | **5.897** |
| FLUX.1-schnell | 0.672 | 0.676 | 5.519 |
| +BoN | 0.869 | 0.704 | 5.497 |
| +RBF | 0.922 | 0.706 | 5.426 |
| +RBF w/ **PRIS** | **0.936** | **0.723** | **5.528** |

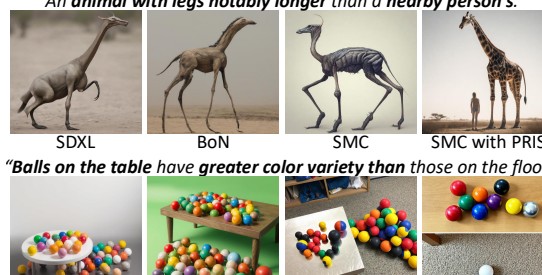

*"An **animal with legs notably longer** than a **nearby person's**."*

SDXL · BoN · SMC · SMC with PRIS

*"**Balls on the table** have **greater color variety than** those on the floor."*

FLUX.1-schnell · BoN · RBF · RBF with PRIS

Figure 6: **Qualitative examples of integrating PRIS with T2I noise-scaling baselines.**

**Effectiveness of iterative prompt revisions with PRIS.** Table 3 evaluates whether iterative revisions, which update prompts based on newly identified failures, provide benefits beyond a single iteration of revision. As shown, iterative revision yields consistent improvements across both given and held-out metrics for prompt adherence while maintaining comparable aesthetic quality. The first update brings a substantial gain, and the second adds further improvements, suggesting that iterative revision progressively strengthens alignment. While multiple revisions yield cumulative gains, the first update already offers a substantial improvement; therefore, we adopt a single refinement step in the main experiments. Moreover, such gains do not appear without PRIS: simply generating more samples with increased compute budget leads to saturated performance on unseen rewards. This highlights that targeted prompt correction is more effective than brute-force visual scaling.

**Integration with visual scaling methods beyond BoN.** PRIS is complementary to prior visual scaling methods that expand the sampling space with fixed prompts. These are noise-based strategies that adjust sampling dynamics, whereas our approach targets prompt-level failures, making it orthogonal to such methods. To validate this complementarity, we integrate PRIS with two established T2I methods, DAS (Kim et al., 2025b) and RBF (Kim et al., 2025a), following their original protocols and evaluating on GenAI-Bench consistent with our main setup. Table 4 and Figure 6 show that integration yields superior alignment on both given and unseen rewards. Notably, whereas RBF often sacrifices aesthetics to improve alignment, our approach improves both simultaneously. Qualitatively, Figure 6 further shows that although DAS and RBF alone struggle on difficult prompts, their integration with PRIS resolves these cases, producing outputs that are both prompt-aligned and visually coherent. Full experimental details, additional examples, and T2V results are provided in Appendix A.

## 4.3 EVALUATING THE VERIFICATION ACCURACY OF EFC

We introduce EFC to address the lack of fine-grained evaluation in text-visual alignment, enabling element-level verification for precise and interpretable assessment. As prompts become more complex, it is critical to check whether all attributes are satisfied rather than relying on a single opaque score. However, widely used human preference datasets (Liu et al., 2025a;b; Wang et al., 2024a; He et al., 2024; Xu et al., 2024) are insufficient: they provide only pairwise judgments and fail to capture whether a single video fully satisfies the prompt. Moreover, they do not reflect inference-time needs, where a verifier must pick the best-aligned sample from a diverse pool of mid-quality outputs. To fill this gap, we construct a benchmark that pairs prompts with both fully aligned (ground-truth) and partially aligned (distractors) visuals, covering varying degrees of completeness and fidelity. Additional dataset details and analyses are provided in Appendix B.

**Constructing the benchmark.** Each prompt is paired with multiple aligned and partially aligned videos, along with tags indicating reasons for misalignment for the partially aligned cases. Prompts are drawn from widely used video model demos and from VBench 2.0 (Zheng et al., 2025), yielding a total of 410 prompts. Candidate videos are generated using diverse state-of-the-art closed- and open-source models (Google DeepMind, 2025; Kuaishou Technology, 2025; Wan et al., 2025). Several human annotators mark a video as aligned if it fully satisfies the prompt and provide explanations when they label it as misaligned. The final label for each video is determined by majority vote.

**Evaluation setup and baselines.** We evaluate EFC and existing verifiers on our benchmark which simulates the inference-time scaling setting. As baselines, we consider widely used learned reward

Table 5: **Quantitative results on verifier accuracy** in selecting GT visual outputs. **Bold** indicates the best results.

| Verifier | Accuracy |
|---|---|
| VisionReward (Xu et al., 2024) | 0.571 |
| UnifiedReward (Wang et al., 2025b) | 0.498 |
| VideoAlign (Liu et al., 2025a) | 0.693 |
| Decomposed binary VQA | 0.700 |
| PRIS (Ours) | **0.763** |

Table 6: **Ablation study of PRIS.** # *d.e.* and # *c.f.* denote the numbers of decomposed elements and common failures, respectively; **bold** indicates the best.

| Task | Prompt Revision | Avg. # *d.e.* | Avg. # *c.f.* | Score |
|---|---|---|---|---|
| T2I | w.o. revision | | - | 0.783 |
| | Per-sample | 3.5 | - | 0.853 |
| | Common-failure | | 0.72 | **0.854** |
| T2V | w.o. revision | | - | 0.711 |
| | Per-sample | 7.3 | - | 0.619 |
| | Common-failure | | 1.46 | **0.754** |

models (Liu et al., 2025a; Xu et al., 2024; Wang et al., 2025b), trained on preference datasets to output a scalar score per video. We then evaluate EFC itself, which performs zero-shot prompt-adherence verification using MLLMs (Bai et al., 2025), along with an ablation that removes its factual correction component. In this ablation, verification is reduced to decomposed visual QA, where each element is judged independently via QA, and the final score is determined by the number of elements marked aligned. For all methods, we select the top-scoring video and evaluate against human annotations.

**Evaluation results.** Table 5 shows that EFC achieves the highest accuracy, significantly surpassing even VideoAlign, the strongest reward model. Moreover, unlike learned reward models, EFC provides fine-grained, interpretable reasoning even without training. It also outperforms decomposed VQA, supporting our design choice of factual correction with text-based verification, consistent with recent findings that text-based measures are more reliable than direct VQA (Fu et al., 2025; Bai et al., 2024).

## 4.4 ABLATION STUDY AND ANALYSIS

**Effect of prompt redesign based on common failure patterns.** We ablate our redesign strategy, which revises prompts by identifying failure patterns shared across the top-$k$ best-aligned samples. This directs corrections toward attributes that are systematically hard to generate (e.g., motion or causal order). As shown in Table 6, common-failure revision consistently outperforms per-sample revision in both T2I and T2V. Notably, per-sample revision in T2V performs worse than no revision, as attempting to fix every failure at once dilutes focus on high-probability misalignments and wastes effort on cases that alternative seeds could easily resolve, ultimately leading to inefficiency. In contrast, our approach effectively focuses corrections on attributes with a high likelihood of recurring failure. Furthermore, the number of common failures (*c.f.* in Table 6) across top-performing seeds supports our motivation (Section 3.1), confirming that different seeds indeed share recurring failure patterns.

**Compute time analysis.** In our experiments, we follow the standard practice of comparing methods under the same NFEs (Ma et al., 2025; Kim et al., 2025a). We also evaluate under matched total wall-clock time (Table 7), including verifiers. Even under this setting, allocating wall-clock time to our framework is more effective than simply increasing NFEs for the generator. Although EFC introduces a modest overhead, mainly from captioning, PRIS achieves substantially larger gains in prompt adherence. These results indicate that directing wall-clock time toward verifier-guided prompt revision is more beneficial than spending the same time on brute-force generation. Please refer to Appendix A for a detailed breakdown of the compute time for verification.

Table 7: **Quantitative evaluation under matched computational time.**

| Task | Method | NFEs (1e3) | Score |
|---|---|---|---|
| T2I | BoN | 4.0 | 0.790 |
| | PRIS | 1.0 | **0.834** |
| T2V | BoN | 4.0 | 0.935 |
| | PRIS | 2.0 | **0.964** |

## 5 CONCLUSION

In this work, we address the overlooked problem of prompt design for inference-time scaling, complementing prior efforts focused solely on expanding the visual search space while keeping prompts fixed. We introduce EFC, a zero-shot verifier that provides fine-grained, interpretable text-visual alignment assessments, and PRIS, an EFC-guided framework that redesigns prompts based on observed common failures across visuals. By reviewing generated outputs to identify recurring misalignment patterns, PRIS adaptively revises the prompt based on the visuals. Across both T2I and T2V generations, our approach delivers significant gains in prompt adherence, demonstrating that jointly scaling prompts and visuals extends scaling behavior beyond visual-only scaling methods. In doing so, our contributions bridge MLLM reasoning and visual generative models, establishing a new direction toward more controllable and high-fidelity text-to-visual generation.

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

# A    ADDITIONAL ANALYSIS

## A.1    COMMON FAILURE PATTERNS

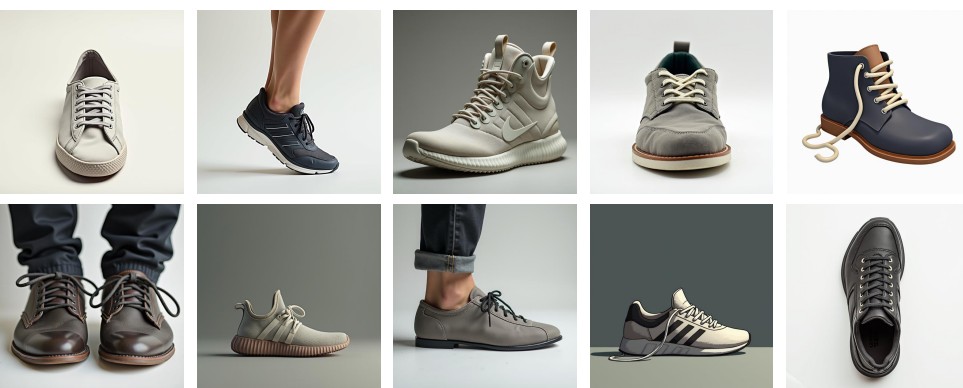

**Original prompt**
: "A shoe with no laces, standing alone"

**Decomposed elements**
1. There is a shoe (core)
2. The shoe has no laces (core)
3. The shoe is standing alone (core)

**Common failure patterns**
: 2. The shoe has no laces

- - - - - - - - - - - - - - - - - - - - - - - - - - - - - - - - - - - - -

**Original prompt**
: "On a wooden table, both the spoons and plates are made of wood, only the fork is not made of wood."

**Decomposed elements**
1. There is wooden dining table (core)
2. There are spoons present (core)
3. There are plates present (core)
4. There is a fork present (core)
5. The spoons are made of wood (core)
6. The plates are made of wood (core)
7. The fork is not made of wood (core)
8. The spoons, plates, and fork are located on the wooden dining table

**Common failure patterns**
: 7. The fork is not made of wood

Figure 7: **Qualitative examples of recurring misalignments** when generating multiple images from a fixed prompt, with decomposed elements and common failures identified by EFC.

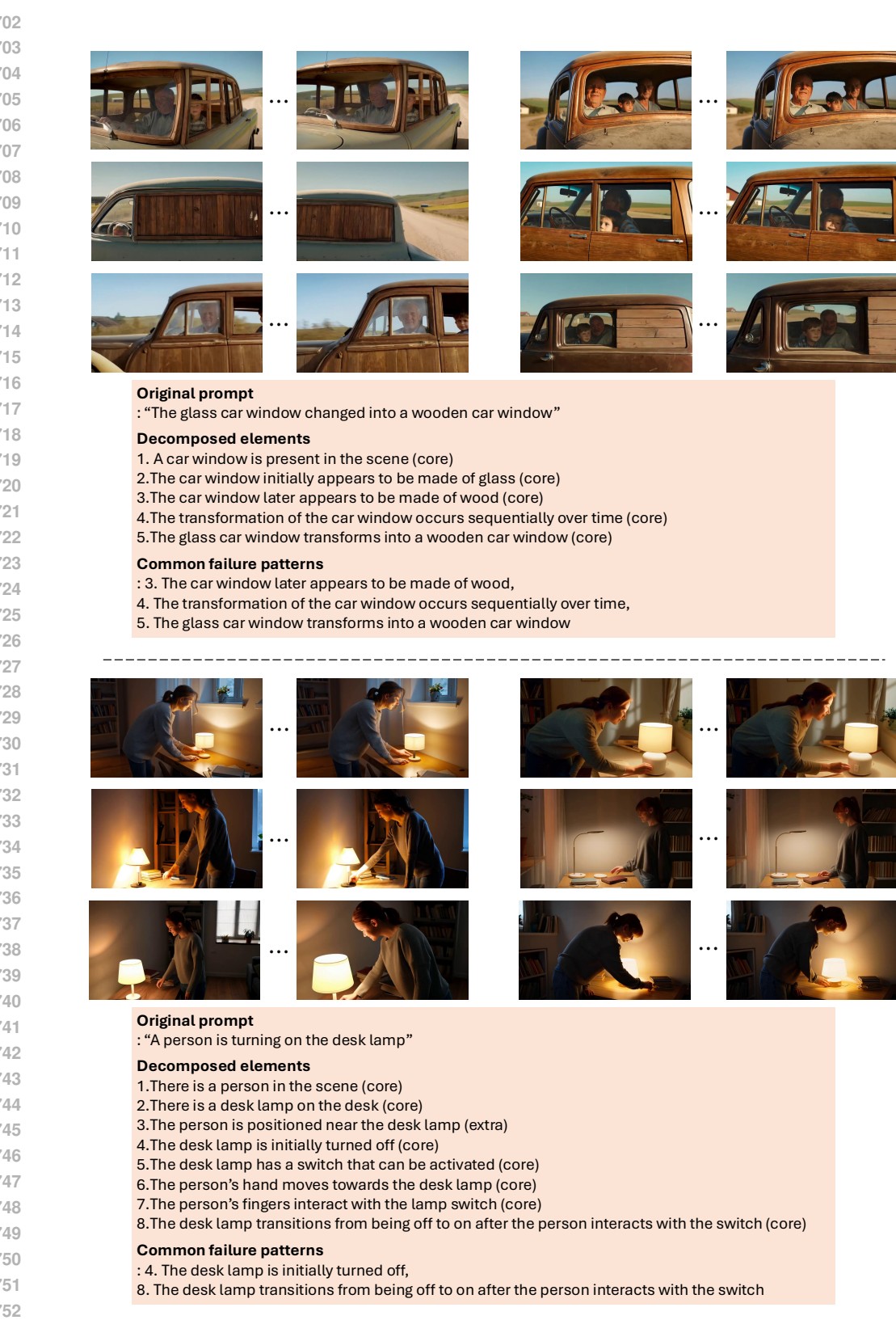

**Original prompt**
: "The glass car window changed into a wooden car window"

**Decomposed elements**
1. A car window is present in the scene (core)
2. The car window initially appears to be made of glass (core)
3. The car window later appears to be made of wood (core)
4. The transformation of the car window occurs sequentially over time (core)
5. The glass car window transforms into a wooden car window (core)

**Common failure patterns**
: 3. The car window later appears to be made of wood,
4. The transformation of the car window occurs sequentially over time,
5. The glass car window transforms into a wooden car window

_________________________________________________________________

**Original prompt**
: "A person is turning on the desk lamp"

**Decomposed elements**
1. There is a person in the scene (core)
2. There is a desk lamp on the desk (core)
3. The person is positioned near the desk lamp (extra)
4. The desk lamp is initially turned off (core)
5. The desk lamp has a switch that can be activated (core)
6. The person's hand moves towards the desk lamp (core)
7. The person's fingers interact with the lamp switch (core)
8. The desk lamp transitions from being off to on after the person interacts with the switch (core)

**Common failure patterns**
: 4. The desk lamp is initially turned off,
8. The desk lamp transitions from being off to on after the person interacts with the switch

Figure 8: **Qualitative examples of recurring misalignments** when generating multiple videos from a fixed prompt, with decomposed elements and common failures identified by EFC. We illustrate the first and the last frame for each generated video.

## A.2 DETAILS OF EFC

We present a detailed overview of the visual verification process in our verifier, EFC. The goal of this process is to provide fine-grained and interpretable feedback on whether each part of a prompt is faithfully realized in the generated visuals. Given a prompt and its corresponding outputs (images or videos), EFC first decomposes the prompt into multiple disjoint semantic elements using a system prompt. For each element, it also constructs an open-ended question, where the element itself serves as the expected answer. Next, EFC verifies the fulfillment of these elements in the generated visuals through factual correction. Instead of relying on visual question answering, our key idea is to perform text-based comparison between the semantic elements and the visuals. To enable this, EFC first extracts captions from the generated visuals and then applies natural language inference (NLI) to classify each element as entailment, neutral, or contradiction. For elements classified as neutral, where captions are missing or insufficient, EFC uses the previously generated open-ended questions, applies NLI again to the corresponding answers, and assigns a final label of either entailment or contradiction. Through this process, EFC not only determines whether the prompt is satisfied, but also pinpoints which parts of the prompt are faithfully represented and which are contradicted, thereby enabling accurate and interpretable fine-grained feedback for generated visuals.

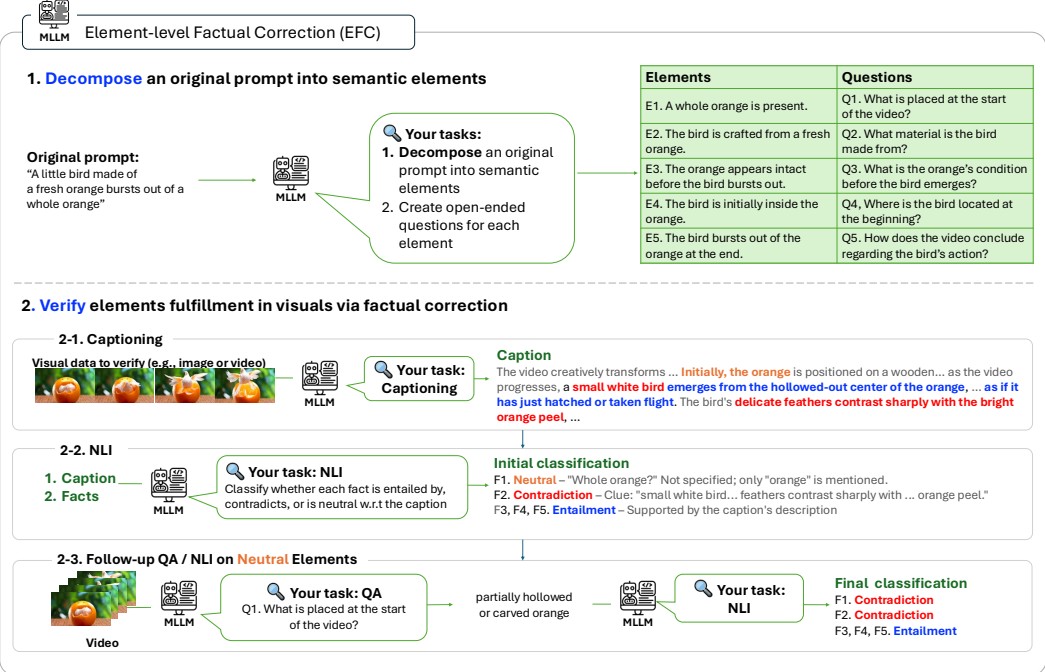

Figure 9: **Illustration of EFC.** The figure illustrates how EFC provides fine-grained, interpretable verification of prompt adherence. It first decomposes the prompt into semantic elements, then generates captions from the visuals, and applies factual correction to classify each element as entailment, neutral, or contradiction. Elements initially labeled neutral (due to missing mentions in the caption) are reevaluated to decide between entailment and contradiction. This design avoids direct QA, leading to more accurate verification.

## A.3 DETAILS ON INTEGRATION BEYOND BON

This section provides additional details on the integration of our framework with visual scaling methods, complementing Section 4.2.

**Experiments on text-to-image generations.** We integrate our approach with two inference-time scaling methods focused on visuals: DAS (Kim et al., 2025b) and RBF (Kim et al., 2025b). Following their original experimental protocols, we use SDXL (Podell et al., 2023) for DAS and Flux.1-schnell (Labs et al., 2025) for RBF. In both settings, we generate a total of 8 samples, divided into two batches of 4. When combined with our method, the first batch of 4 samples is generated, the prompt is revised, and another 4 samples are generated, ensuring that the total number of function evaluations remains equivalent.

Figure 10: **Qualitative artifact results with RBF.** RBF alone often generates visuals where the prompt text is directly rendered on the image due to reward over-optimization, whereas combining RBF with our method substantially alleviates this issue.

In addition to Table 4 and Figure 4 in the main manuscript, Figures 11 and 12 demonstrate that our integrated results achieve substantially better prompt adherence than visual scaling alone, for DAS and RBF, respectively. This indicates that advanced visual scaling methods can be further enhanced when combined with scaled prompts.

It is also noteworthy that scaling visuals alone often leads to undesired outcomes caused by reward over-optimization (see Figure 10). In such cases, the model may even render the textual prompt itself, since these images achieve artificially high reward scores. For example, Figure 10 shows that RBF frequently generates images where the prompt text is printed directly. By contrast, our method mitigates this issue: the revised prompt guides the generator, while the original prompt is used only for the reward signal. This separation effectively reduces over-optimization artifacts and yields more faithful generations, even when PRIS is combined with RBF.

Figure 11: **Qualitative comparisons when integrating our method with SMC** under the same compute budget. Our approach more faithfully follows the prompt, effectively enabling SMC to scale visuals.

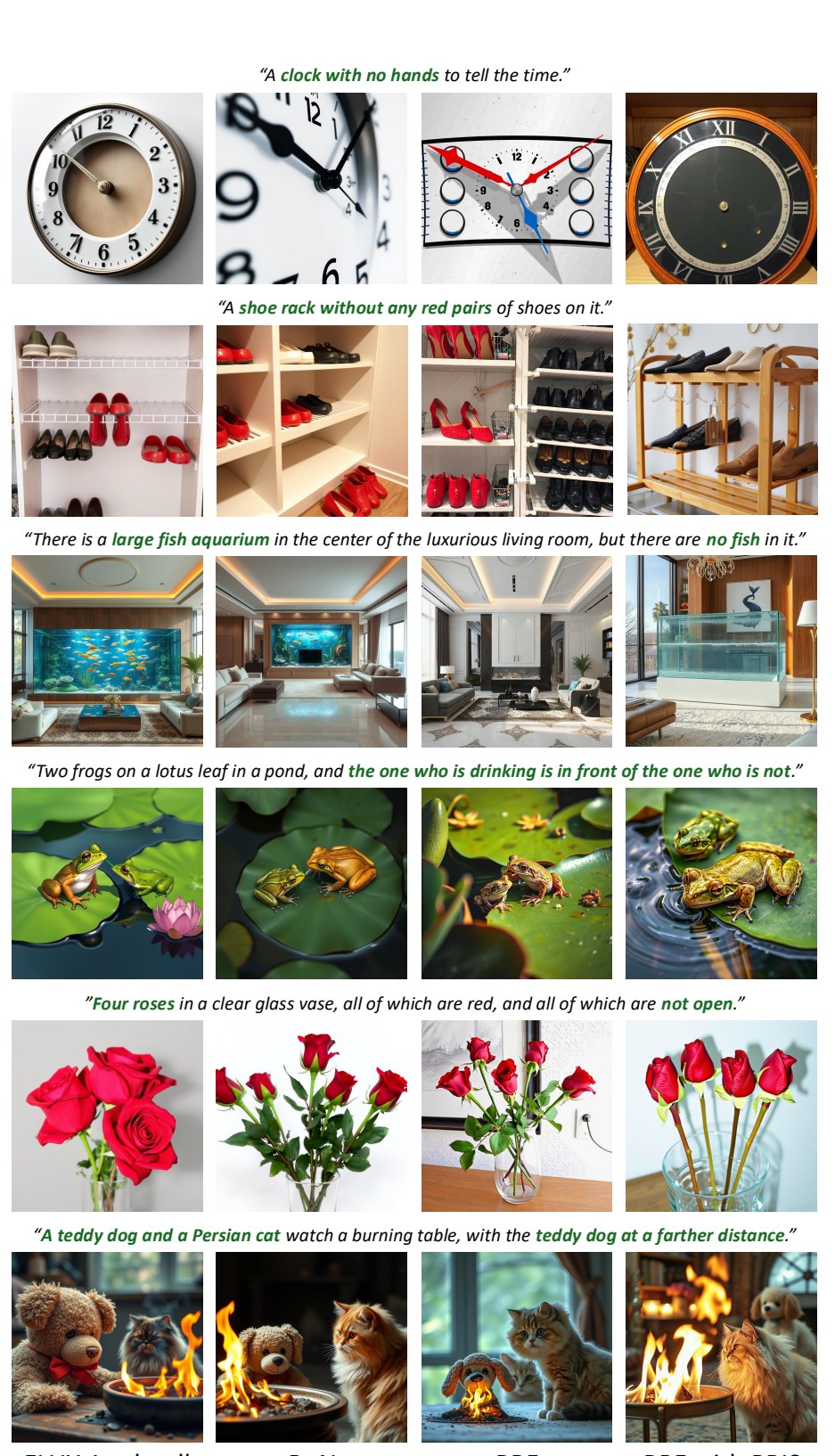

Figure 12: **Qualitative comparisons of RBF integrated with our method** under the same compute budget. Our method adheres more closely to the prompt and further improves RBF's visual scaling.

**Experiments on text-to-video generations.** We integrate our approach with EvoSearch (He et al., 2025), following its original setup on Wan2.1-1.3B. EvoSearch uses an evolution schedule of [5, 20, 30, 45] and a population schedule of [10, 5, 5, 5], totaling 2,000 NFEs. For integration, we first generate 10 samples with 50 steps (1,000 NFEs), then allocate the remaining 940 NFEs with [5, 30] for the evolution schedule and [5, 4] for the population schedule, resulting in 60 fewer NFEs than EvoSearch. As in the main manuscript, we evaluate on VBench2.0 with VideoAlign as the guiding reward.

Table 8 and Figure 13 present the quantitative and qualitative results. Unlike EvoSearch, which was evaluated on relatively simple prompts, our experiments employ more complex ones. In this setting, EvoSearch scores degrade after scaling, suggesting limited generalization to the unseen reward of VBench2.0. By contrast, when integrated with our method, it achieves improved average scores on VBench2.0.

Table 8: **Quantitative T2V results on VBench2.0, comparing EvoSearch alone with EvoSearch integrated with PRIS**. EvoSearch fails to generalize to unseen rewards, whereas integration with PRIS improves performance.

| Method | Motion Rationality | Motion Order Understanding | Dynamic Attribute | Average |
|---|---|---|---|---|
| Wan2.1-1.3B | 38.10 | **52.87** | 46.67 | 45.88 |
| EvoSearch | 32.14 | 51.72 | 43.33 | 42.20 ↓ −3.68 |
| EvoSearch + PRIS | **53.57** | 48.28 | **60.00** | **53.95** ↑ +8.07 |

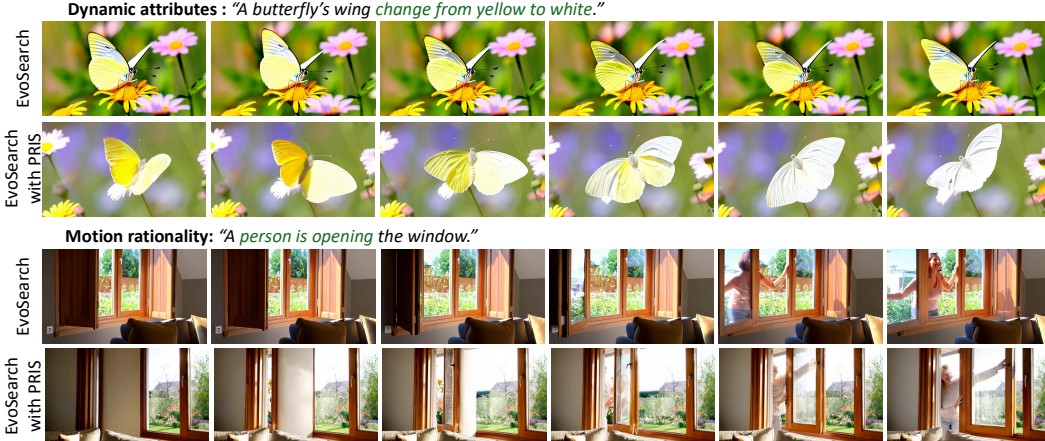

Figure 13: **Qualitative examples comparing EvoSearch and EvoSearch+PRIS**. In the first case, EvoSearch fails to change the butterfly's wing color despite scaling, whereas our method succeeds. In the second case, EvoSearch depicts the window as already open before the person attempts to open it, while our method correctly shows the window opening as the person reaches out.

### A.4 DETAILED COMPUTATIONAL TIME ANALYSIS

In this section, we provide a detailed breakdown of verification and generation time, complementing Section 4.4. All measurements are conducted on a single NVIDIA H100 80GB GPU. For images, generating a single sample resolution $(1024, 1024)$ with Flux.1-dev takes on average 13 seconds, while verification with our verifier, EFC, requires 41 seconds. This implies that each verification is computationally equivalent to generating approximately three additional images. To balance this overhead, we set the number of function evaluations (NFE) to 4000 for BoN and 1000 for our method, corresponding to 40 and 10 images, respectively (with 50 sampling steps and classifier-free guidance). For videos, generating an 81-frame sequence at resolution $(480, 832)$ with Wan2.1-1.3B requires 105 seconds on average, while verification takes 100 seconds, approximately equivalent to one additional video generation. Accordingly, we set the NFE to 4000 for BoN (40 videos) and 2000 for our method (20 videos), again under 50 sampling steps with classifier-free guidance.

Our verifier is intentionally built on a pretrained MLLM without task-specific optimization, demonstrating that strong results can be achieved without additional training. Nonetheless, fine-tuning the base MLLM remains a promising direction for reducing verification time and improving efficiency.

## A.5    PROMPT TRANSFERABILITY AND FUTURE WORK

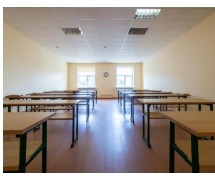 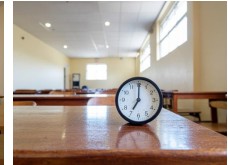

**Original Prompt**: In a classroom, the clock is not on the wall

**Revised Prompt**: In a classroom, **the clock is placed on a polished wooden desk**, its round face softly illuminated, while the walls remain unadorned, free of any other timepieces.

**Original Prompt**          **Revised Prompt**

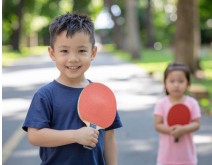 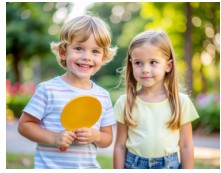

**Original Prompt**: A little boy with a ping pong paddle looks more excited than a little girl without one.
**Revised Prompt**: A young boy holding a bright yellow ping pong paddle beams with **wide eyes and an open smile**, while nearby, a **calm little girl** gazes at him with a curious expression, **her hands resting by her side.**

**Original Prompt**          **Revised Prompt**

Figure 14: **Qualitative example of prompt transferability**. Prompts revised for Flux1.dev are applied to Firefly Image 4 Ultra. By clarifying vague instructions, specifying object presence and absence, and reinforcing contextual cues, the revised prompts yield visuals with stronger adherence compared to those generated from the original prompts.

We observe that our revised prompts are not only effective for the original generator but also transferable to other models, demonstrating their generalizability. This stems from the fact that our revisions resolve ambiguities in the original prompts, making them more precise and robust. Although different generators may specialize in certain aspects, such as producing fine-grained details or maintaining object counts, they often exhibit overlapping weaknesses. Addressing these weaknesses through prompt revision thus benefits multiple models simultaneously.

Figure 14 illustrates this transferability. The prompts originally revised for Flux1.dev are successfully applied to Firefly Image 4 Ultra. For example, the revised prompts clarify vague or underspecified instructions (e.g., replacing "not on the wall" with "the clock is placed on a polished wooden desk"), making object presence and absence explicit (e.g., reformulating "the girl is without a ping pong paddle" into "her hands resting by her side"), and reinforcing contextual cues.

These findings suggest a promising research direction: fine-tuning LLMs or other prompt-rewriting systems on pairs of naïve user-provided prompts and failure-focused revisions. By learning systematic transformations from short, underspecified, and loosely written prompts into precise, detailed, and effective ones, rather than relying on random expansions, such models could reduce verification costs and inference-time overhead, accelerating the discovery of high-quality prompts from the outset.

## A.6    FUTURE WORK AND LIMITATIONS.

We believe our benchmark opens a new avenue for evaluating verifiers at the attribute level. We also observe that prompts refined on one model often generalize well to others (see Appendix A). Building on this insight, future work could explore fine-tuning LLMs or other prompt-rewriting models using pairs of randomly expanded and failure-focused prompts as training assets, which reduce verification overhead and inference-time costs, enabling more efficient discovery of effective prompts from the start.

## A.7    THE USE OF LARGE LANGUAGE MODELS

We only utilize Large Language Models (LLMs) to aid and polish our writing. However, they are not used to the extent that they could be considered contributors or sources of research ideas.

# B    BENCHMARK CONSTRUCTION AND EVALUATIONS

## B.1    BENCHMARK CATEGORY

**Details about benchmark constructions.** Existing visual evaluation datasets are mostly limited to human-preference annotations. While useful for coarse quality assessment, such datasets are insufficient for our focus: selecting the best-aligned videos from among multiple misaligned candidates, which lies at the core of inference-time scaling. To address this limitation, we construct a new benchmark explicitly tailored for inference-time scaling and use it to evaluate our verifier, its ablations, and existing baselines. Beyond serving as a testbed for our study, this benchmark also provides a valuable resource for future research on visual prompt-adherence verification.

In our benchmark, each prompt is paired with multiple generated videos, with at least one ground-truth (GT) aligned reference and others containing slight misalignments, thereby forming a mid-quality candidate pool. In total, the benchmark comprises 410 prompts. We collect prompts showcased in demos of both popular open-source (Wan et al., 2025) and closed-source video models (Google DeepMind, 2025; Kuaishou Technology, 2025), and categorize them into two broad groups: motion (120 prompts) and physics (144 prompts). To further enrich the evaluation, we also adopt prompts from VBench 2.0, spanning three fine-grained motion-related categories: dynamic attributes (47 prompts), motion order (68 prompts), and motion rationality (31 prompts). For each prompt, we generate videos using multiple text-to-video models (xx, xxx) as well as image-to-video models, ensuring the inclusion of both GT-aligned and misaligned outputs. Each video is independently annotated by three human evaluators as GT or non-GT, and the final label is assigned by majority vote.

**Detailed analysis of verifiers on our benchmark per category.** In addition to the overall accuracy reported in Table 5 of the main manuscript, we present per-category accuracy in Table 9. As the results show, EFC consistently achieves the highest accuracy across all categories. Compared to the decomposed binary VQA baseline, which shares our decomposition strategy but replaces our text-to-text verification with binary VQA, EFC yields a substantial performance gain, underscoring the advantage of our text-based approach over visual QA methods. When compared to learned reward models (i.e., MLLM-based verifiers fine-tuned on human-preference datasets), including VideoAlign (the strongest among them and used as our tie-breaker), EFC still maintains a significant lead. Notably, it achieves this performance without any additional training on preference datasets, but rather through a systematic zero-shot verification process. Furthermore, we attribute this gap to the fact that reward models are typically trained on human-preference data, where subtle aspects such as frame quality, motion smoothness, or stylistic biases often dominate judgments, even when they are not directly related to prompt adherence. In contrast, EFC focuses explicitly on verifying semantic alignment with the prompt, making it both more accurate and interpretable.

Table 9: **Quantitative results of verifier accuracy per prompt category on our constructed dataset. Bold** indicates the best result.

| Method | Motion | Physics | Dynamic Attributes | Motion Rationality | Motion Order Understanding | Average |
|---|---|---|---|---|---|---|
| VisionReward (Xu et al., 2024) | 0.650 | 0.569 | 0.319 | 0.662 | 0.452 | 0.571 |
| UnifiedReward (Wang et al., 2025b) | 0.492 | 0.507 | 0.298 | 0.588 | 0.581 | 0.498 |
| VideoAlign (Liu et al., 2025a) | **0.792** | 0.660 | 0.511 | 0.794 | 0.516 | 0.693 |
| Decomposed binary VQA | 0.733 | 0.667 | 0.617 | 0.809 | 0.613 | 0.700 |
| PRIS (Ours) | **0.792** | **0.764** | **0.638** | **0.838** | **0.677** | **0.763** |

While our study focuses on prompt-adherence verification, we believe that our verification framework can be extended to other important axes of evaluation, such as motion quality, NSFW filtering, and bias detection, by replacing prompt decomposition with task-specific decomposition strategies. This flexibility offers promising directions for future research.

# C  EXPERIMENTS DETAILS

## C.1  DETAILED SETUP

For GenAI-Bench, since many prompts within the same categories (e.g., counting, differentiation, comparison, negation, universal) are similar but differ only in objects, we randomly subsample 20% to reduce redundancy. For selecting $k$, we set $k = N//4$, as $N//2$ samples are first generated for review before prompt revision, and half of them are used as top-performing seeds.

## C.2  BASE MODEL SELECTION

To ensure that our study focuses on the effect of prompt redesign in inference-time scaling, we first measure the degree of prompt adherence across candidate leading open-source video models such as Wan, LTX, and Hunyuan. This step is necessary because if a model fails to follow the prompt at all, there is little need to apply prompt redesign. Specifically, we compute the text embedding similarity between the original prompt and the generated video caption. We use Qwen-32B for captioning and employ the SentenceTransformer model (`intfloat/e5-mistral-7b-instruct`) to measure embedding similarity. We present the similarity score in Table 10.

Table 10: **Quantitative results of prompt adherence** across different text-to-video models, used to exclude base models with poor alignment and retain only those with acceptable adherence.

| Metric | Method | Motion Rationality | Motion Order Understanding | Dynamic Attribute | **Average** |
|---|---|---|---|---|---|
| VideoAlign | LTX | 0.764 | -0.153 | -0.977 | -0.122 |
| | Hunyuan | 0.904 | 0.212 | -0.775 | 0.114 |
| | Wan | **1.475** | **0.940** | **-0.397** | **0.673** |
| Text Similarity | LTX | 0.635 | 0.642 | 0.600 | 0.626 |
| | Hunyuan | 0.678 | 0.671 | 0.616 | 0.655 |
| | Wan | **0.717** | **0.702** | **0.631** | **0.683** |

Based on this analysis, we selected Wan as our primary video model, since it demonstrates a reasonable level of prompt adherence while leaving room for improvement through verification and redesign. In contrast, models such as LTX and Hunyuan were excluded, as their low adherence made them unsuitable for evaluating prompt redesign at inference-time scaling, particularly on complex prompts in VBench2.0 that involve status changes or multiple consecutive events within a single video.

# D ADDITIONAL QUALITATIVE EXPERIMENTAL RESULTS

## D.1 TEXT-TO-IMAGE GENERATION

We provide additional qualitative results beyond Figure 3, demonstrating that our prompt redesign improves coherence of the final visual outputs under the same NFE budget (2000, as in the main experiments). Specifically, our method excels on prompt sets with ambiguous attributes, numerical details, or subtle constraints (e.g., "without," "greater variety"), effectively elaborating them into more faithful visual realizations than baselines. We compare the top-scoring outputs generated from the original GenAI-Bench prompts (Table 15) with those from their expanded variants (marked with *, Table 16).

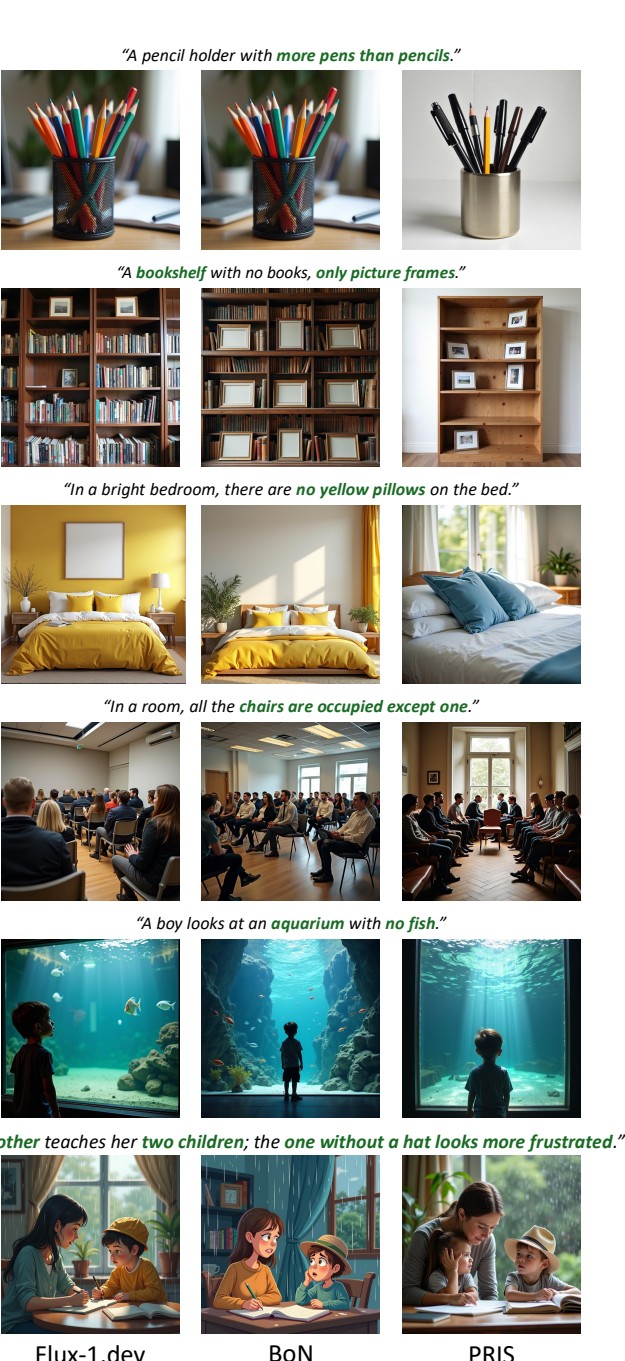

Flux-1.dev          BoN          PRIS

Figure 15: **Qualitative comparisons on T2I generation** where visual generation is (initially) conditioned on the original prompts.

For complex prompts, our joint scaling of visuals and prompts more faithfully preserves the intended semantics within the same compute budget by adaptively revising the prompts, unlike standard prompt expansion that cannot target and identify the most difficult semantic elements.

*"**Two excited elephants** to the right of a **lost giraffe**."*

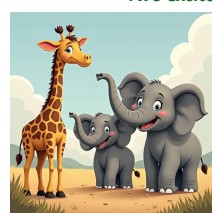 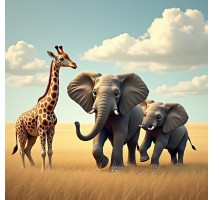 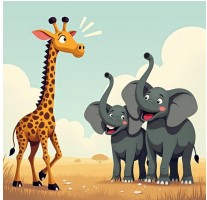

*"A monkey with a backpack is **jumping from one smaller three to another larger tree**."*

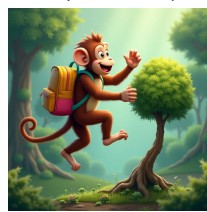 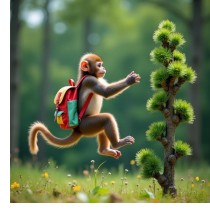 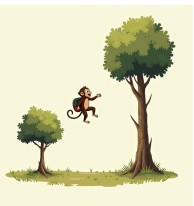

*"A **farm** with a barn that **does not shelter any sheep**."*

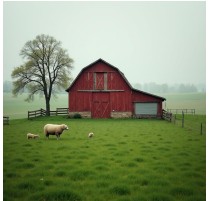 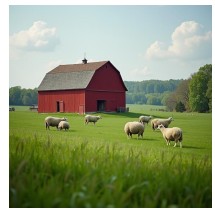 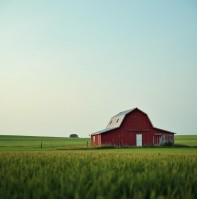

*"A **bed without the usual cat** sleeping on it."*

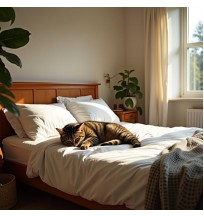 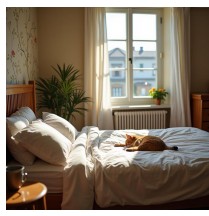 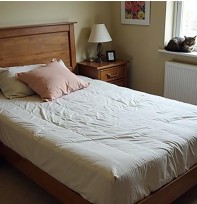

*"The **balls on the table have a greater variety of colors** than the ones on the floor."*

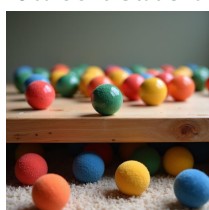 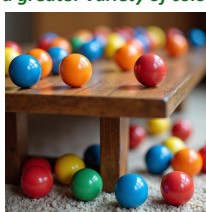 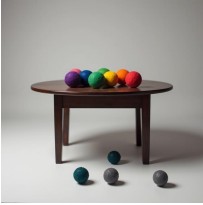

*"The two lay in bed, the **long-haired one asleep**, the **short-haired one still awake**."*

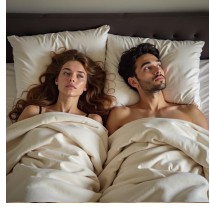 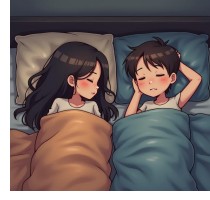 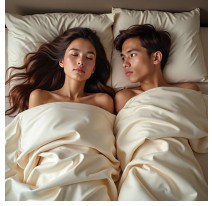

| Flux-1.dev* | BoN* | PRIS* |

Figure 16: **Qualitative comparisons on T2I geneation** where visual generation is (initially) conditioned on standard prompt expansion.

## D.2 TEXT-TO-VIDEO GENERATION

In addition to Figure 4, we present additional qualitative top-scoring examples in Figure 17. As shown, our method more faithfully follows the intent of the original prompt. The final top-scoring visuals generated with our PRIS demonstrate significantly stronger prompt adherence compared to baselines. Specifically, BoN often misses key events or produces unnatural temporal order. For example, it may depict only a single motion (e.g., morphing without differentiating "cleaning the kitchen" in the 1st visual) or assign different motions to different people (in the 4th visual). BoN also frequently fails to capture dynamic changes, generating only static states (3rd and 6th visuals). Furthermore, BoN often does not correctly realize sequential actions, such as repeatedly attempting to break chocolate pieces, whereas our method generates coherent sequences where the person both attempts the action and displays the broken pieces (5th visual).

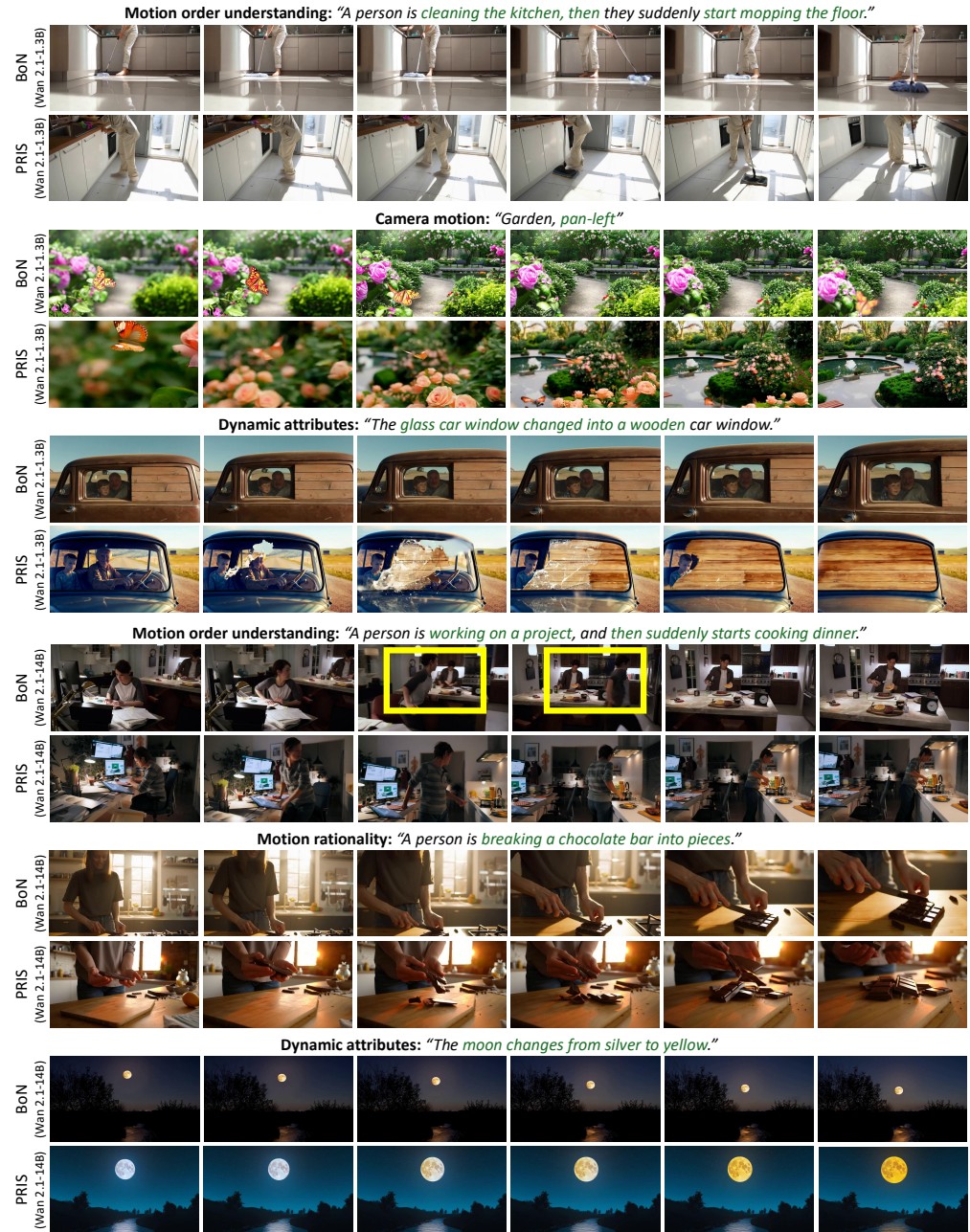

Figure 17: **Qualitative comparisons on T2V generation** where visual generation is (initially) conditioned on standard prompt expansion, with Wan2.1-1.3B (top) and Wan2.1-14B (bottom).

### D.3 MORE VISUALIZATIONS

We include an HTML file to the attached zip file. To explore the generated visuals and comparisons with baselines alongside their corresponding prompts, please open `visuals/index.html` in a Chrome browser (This file is located in the `visuals` directory within the attached zip file). This visualizes the generated visuals, including images and videos, in the `visuals/resources` folder.

