# OpenReview forum: "Rethinking Prompt Design for Inference-time Scaling in Text-to-Visual Generation"
_ICLR.cc/2026/Conference — ICLR 2026 Conference Withdrawn Submission_

### Official Review · Reviewer_DA4Y · 2025-10-16

**Soundness:** 2
**Presentation:** 3
**Contribution:** 2
**Rating:** 4
**Confidence:** 4

**Summary:**

This paper studies the test-time prompt refinement in text-to-image/video generation. The authors claimed that existing methods improve results mainly by allocating additional compute (e.g., generating more samples using more seeds or sampling steps), with the text prompt remains fixed. And this method quickly leads to a quality plateau. Also for the textual prompts, the author argued that they're crucial for guiding the generation.

To address this, the author proposed a framework consists of two main components:
1. EFC: A fine-grained verifier that checks how well each part of a prompt (object, attribute, action, relation, etc.) appears in the generated visual. It decomposes the prompt into semantic elements, compares them with visual captions using entailment / contradiction / neutral, and identifies which parts are missing or wrong.
2. PRIS:
a. Generate multiple visuals for a given prompt.
b. Use EFC to verify which elements are satisfied or failed.
c. Identify common failure patterns across samples.
d. Revise the prompt to emphasize missing elements while keeping the original intent.

The experiments shows the effectiveness of EFC + PRIS, yielding better results than BoN.

**Strengths:**

1. The writing and illustrations of the paper is good and easy to follow. The task settings is clear. The appendix is rich.
2. The proposed method is intuitive and reasonable to the reviewer.
3. This paper focuses on an interesting question in the field of visual generation.

**Weaknesses:**

1. The method is good overall. But the reviewer still concerns the performance gained compared with prior "prompt-refine" methods severely. As there are a lot of existing methods focusing on test-time "prompt-refine" after "Design Guidelines for Prompt Engineering Text-to-Image Generative Models [CHI 2022]", like:
a. Optimizing Prompts for Text-to-Image Generation [NeurIPS 2023]
b. From Reflection to Perfection: Scaling Inference-Time Optimization for Text-to-Image Diffusion Models via Reflection Tuning [ICCV 2025].
c. VPA: Fully Test-Time Visual Prompt Adaptation [ACM MM 2023]
I'm sure you can find a bunch of papers from the citation list of Design Guidelines for Prompt Engineering Text-to-Image Generative Models [CHI 2022].
Thus, more comparisons between your prompt design method and previous methods should be conducted.

2. The author assumed that "prior approaches mainly scale the visual generation process (e.g., increasing sampling steps or seeds), but this quickly leads to a quality plateau. We argue that this limitation arises because the prompt, crucial for guiding generation, is kept fixed.". The reviewer highly doubt it. Not all previous works are passively waiting for a high-scoring sample, whithout adaptively changing some "conditions" (e.g., inputs, weights, etc.) that can affect the generation. The reviewer consider it'd be much better to discuss the drawbacks of previous prompt design methods with yours.

3. Also, similiar to point 1, more reward models should be compared, like:
a.  ImageReward: Learning and Evaluating Human Preferences for Text-to-image Generation[NeurIPS 2023]
b. Pick-a-Pic: An Open Dataset of User Preferences for Text-to-Image Generation [NeurIPS 2023]
c. Better Aligning Text-to-Image Models with Human Preference. [ICCV 2023]
The authors should be more responsible to do a good survey instead of the reviewer.

**Questions:**

1. What's the essential difference between your method and BoN? To the best of the reviewer's knowledge, the only difference is the "learn from recurring failure patterns" as defined in EFC. However, why this additional step can boost the performance clearly and worth the additional computation time it takes? Isn't picking out common failure cases the reverse BoN? The review still thinks there is no essential difference between your method and BoN, alghough different methods can be similar in the high-level idea. More discussion of the implementation of the "BoN" in your experiment and the difference between it and your method is encouraged.

---

### Official Review · Reviewer_weS4 · 2025-10-30

**Soundness:** 3
**Presentation:** 2
**Contribution:** 2
**Rating:** 4
**Confidence:** 4

**Summary:**

This paper introduces a novel prompt redesign strategy, PRIS (Prompt Redesign for Inference-time Scaling), which aims to achieve precise alignment between user intent and the generated visuals. PRIS accomplishes this by utilizing the EFC (Element-level Factual Correction) verifier to analyze multiple generated visual candidates, diagnose recurring common failures across these outputs, and subsequently revise the original prompt to reinforce the under-addressed aspects.

**Strengths:**

1. The idea of using an MLLM to analyze the common failures across multiple, distinct visual generations and then revising the prompt based on the common failures is interesting.
2. The paper demonstrates strong performance, achieving significant gains.

**Weaknesses:**

1. This paper focuses primarily on common failures while overlooking other potential failure cases, suggesting that the final outputs may still suffer from misalignment issues.
2. Several statements and procedures lack sufficient detail or contain inaccuracies. For example:
- The claim in Line 52 that prior methods are limited because they “operate solely in the text domain” seems unfair, as your method also operates in the text domain (i.e., by redesigning the prompt).
- The relationship between NFE, N, denoising step, CFG is unclear. When NFE = 1000, 2000, or 3000, it is not specified which specific factors are being changed.
- The paper does not specify which model or rules are used to obtain the revised prompt based on EFC’s common failure.
3. While the presented strategy of diagnosing common failures across multiple samples is clever and yields strong results,the novelty feels somewhat insufficient.

**Questions:**

See Weaknesses

---

### Official Review · Reviewer_Y9cR · 2025-11-08

**Soundness:** 3
**Presentation:** 3
**Contribution:** 3
**Rating:** 8
**Confidence:** 3

**Summary:**

The paper proposes a novel approach to improve the alignment of input text and visual output in T2I (text to image) and T2V (text to video) generation during inference time by identifying the common failures of output samples and redesigning the input prompt. At the core of its method, the paper proposes a method called Element-level Factual Correction (EFC) that decomposes the input prompt into the key semantic elements of the input prompt. The idea is then to generate M samples using the original input prompt, to identify the top-k output samples and their recurring failures, to revise the prompt elements that are responsible for these failures, and to then re-generate the samples using the same seeds using the revised prompt. While being methodologically simple, all design steps are well justified. The paper presents convincing results and provides a thorough evaluation both for image generation as well as for video generation and can show its effectiveness both quantitatively and qualitatively.

Although I am not an expert in this particular field, I believe that this paper and its findings are of great value to the ICLR community. I am therefore leaning positively and would argue to accept the paper.

**Strengths:**

The paper tackles an important and interesting problem and motivates it very well. It further gives a good overview of the state of the art. I also had the feeling that all design steps were well justified. The results are convincing and clearly show the effectiveness of the proposed method. The overall presentation of the paper is very clear and allows fluent reading. The proposed method is thoroughly evaluated w.r.t. various different perspectives (e.g., fixed compute budget, integration with visual scaling algorithms).

**Weaknesses:**

My main concern about the paper is mostly about its simplicity and the way some prompts are formulated to make the baseline methods look bad. I had the feeling that many results where the baseline methods looked bad in comparison could be easily resolved by taking the input prompt and asking an LLM to reformulate negations. Many results shown in the paper have prompts like "no laces", "fork is not wooden", "not wearing a helmet" etc. By just throwing out negations early-on could probably solve many issues in the generated images and videos.


Minor comments

In Section 3.3, the caption of Figure 4 should probably be "Qualitative comparisons..." instead of quantitative.

In the Appendix in Section A.2, Figure 9 is never referenced.

In the Appendix in Section B.1, line 1098, there should probably be a listing of examples text-to-video models instead of the current placeholder "(xx, xxx).

**Questions:**

I wonder how a comparison to some of the baseline methods would look like if the input prompt would really just do a very simple preprocessing step to remove all negations and vague formulations. How would some of the results then look like, e.g. the shoe without laces and the non-wooden fork? I guess ChatGPT could very easily change the prompt to something like "a loafer standing alone" or "a work made of metal or plastic". How many problems would this solve?

---

### Official Review · Reviewer_bHkF · 2025-11-11

**Soundness:** 2
**Presentation:** 3
**Contribution:** 1
**Rating:** 2
**Confidence:** 2

**Summary:**

This paper proposes PRIS (Prompt Redesign for Inference-time Scaling), a framework to improve text-to-visual generation. The core idea is to move beyond fixed prompts during inference-time scaling (like Best-of-N). PRIS first generates an initial batch of visuals, then uses a new verifier called EFC (Element-level Factual Correction) to analyze them. EFC decomposes the prompt into semantic elements, identifies "common failure patterns" across the generated batch, and revises the prompt to specifically address these failures. Finally, it regenerates new visuals using this revised prompt and the seeds from the best-performing initial samples. The authors present this as a method for "jointly scaling prompts and visuals" and report significant improvements on T2I and T2V benchmarks.

**Strengths:**

- Strong Empirical Results: The paper shows impressive quantitative gains, such as a +15% improvement on VBench 2.0 and consistent outperformance against BoN and other baselines (Table 1, 2, 4).
- Clear Problem Formulation: The paper correctly identifies a key limitation of existing inference-time scaling methods: they scale visuals (e.g., sampling steps, seeds) but keep the prompt fixed, which leads to a quality plateau.

**Weaknesses:**

- Critical Lack of Novelty: As detailed under "Contribution," the paper's core idea is not new. The PRIS framework (Generate -> Verify -> Revise -> Regenerate) is functionally identical to the "reflection and guidance" or "verify and reinforce" loops proposed in prior CoT-based generation work, such as Guo et al. (2025) and Jiang et al. (2025).
- Incremental Contribution: The paper's attempt to distinguish itself by using "off-the-shelf MLLMs" instead of "unified models"  is a minor implementation detail, not a novel conceptual contribution.
- Contradictory Method: The EFC verifier's methodology is contradictory. It claims to "mitigate... affirmative bias" of VQA by using a text-to-text NLI approach, but then explicitly falls back to a "Follow-up QA" step  for "neutral" elements, re-introducing the very method it claims to be superior to.

**Questions:**

- The core idea of "verify and reinforce" or "reflection and guidance" is central to works like Guo et al. (2025) and Jiang et al. (2025), which you cite. Can you please articulate, beyond minor implementation details (e.g., "off-the-shelf MLLMs"), what the fundamental conceptual novelty of PRIS is?
- Your EFC verifier claims to mitigate VQA bias by using a "text-to-text comparison". However, your pipeline (Fig 2a, Sec 3.2) explicitly includes a "Follow-up QA" step  for neutral elements. How do you reconcile this with your claim of avoiding VQA-based methods?

---

### Note · Authors · 2025-11-13

**Comment:**

We sincerely appreciate the reviewers’ time and constructive feedback :)

**Withdrawal Confirmation:**

I have read and agree with the venue's withdrawal policy on behalf of myself and my co-authors.